# Construction and Application of Functional Brain Network Based on Entropy

**DOI:** 10.3390/e22111234

**Published:** 2020-10-30

**Authors:** Lingyun Zhang, Taorong Qiu, Zhiqiang Lin, Shuli Zou, Xiaoming Bai

**Affiliations:** Department of Computer, Nanchang University, Nanchang 330029, China; Z296207126@gmail.com (L.Z.); 401030918054@email.ncu.end.cn (Z.L.); zsl@ncu.edu.cn (S.Z.); baixiaoming@ncu.edu.cn (X.B.)

**Keywords:** functional brain network, fuzzy entropy, fatigue driving

## Abstract

Functional brain network (FBN) is an intuitive expression of the dynamic neural activity interaction between different neurons, neuron clusters, or cerebral cortex regions. It can characterize the brain network topology and dynamic properties. The method of building an FBN to characterize the features of the brain network accurately and effectively is a challenging subject. Entropy can effectively describe the complexity, non-linearity, and uncertainty of electroencephalogram (EEG) signals. As a relatively new research direction, the research of the FBN construction method based on EEG data of fatigue driving has broad prospects. Therefore, it is of great significance to study the entropy-based FBN construction. We focus on selecting appropriate entropy features to characterize EEG signals and construct an FBN. On the real data set of fatigue driving, FBN models based on different entropies are constructed to identify the state of fatigue driving. Through analyzing network measurement indicators, the experiment shows that the FBN model based on fuzzy entropy can achieve excellent classification recognition rate and good classification stability. In addition, when compared with the other model based on the same data set, our model could obtain a higher accuracy and more stable classification results even if the length of the intercepted EEG signal is different.

## 1. Introduction

A functional brain network (FBN) is the characterization of the brain network topology and dynamic properties [1]. Different brain network typologies and characterizations can be obtained according to different construction methods based on the same electroencephalogram (EEG) data set, which have a profound impact on the research of detection and recognition of different brain states. Therefore, the research of the FBN construction method is very important. Meier et al. [2] constructed the union of shortest path trees (USPT) as a new topology for the FBN, which can be uniquely defined. The new concept of the USPT of the FBN also allows interesting interpretation and may represent the “highway” of the brain after interpreting the link weights of the FBN. Kabbara et al. [3] used EEG source connectivity computed in different frequency bands to reconstruct functional brain networks based on the EEG data of 20 participants (10 Alzheimer’s disease patients and 10 healthy individuals) in a resting state network, and used graph theoretical analyses to evaluate the differences between both groups. The results revealed that Alzheimer’s disease networks, compared to networks of age-matched healthy controls, are characterized by lower global information processing (integration) and higher local information processing (segregation). Zou et al. [4] proposed a functional brain network construction method based on the combination of shortest path tree (CSPT) (abbreviated as CSP-FBN), and compared with the classification accuracy of the same frequency band (beta-band). They found that the fatigue state recognition functional brain network constructed by combining the shortest path tree was superior to functional brain networks constructed by other methods, with an accuracy of 99.17%, and found that Fz, F4, Fc3, Fcz, Fc4, C3, Cz4, Cp3, Cpz, Cp4, P3, Pz, and P4 are important electrodes for identifying the state of fatigue driving, reflecting that the correct central area of the central parietal lobe of the brain is closely related to fatigue driving. Conrin et al. [5] used the PACE algorithm, which permits a dual formulation. The positive and negative edges of the FBN were all given the equivalent modular structure of the connection group. The clinical relevance of the relationship between gender differences and age were more thoroughly examined, in which they found the supporting infrastructure framework and conceptualized the functional connector. Zhao et al. [6] applied an FBN to construct a fatigue recognition model based on EEG data and graph theory. It was observed that the coherence of the frontal, central, and temporal lobe of the brain was significantly improved when using this method, and in addition the clustering coefficients and length of the character path β and α are significantly increased. These valuable research experiences confirmed that different FBN methods will lead to different brain network topology, which has a certain impact on the classification and recognition of different brain activity states.

Entropy can effectively describe the complexity, non-linearity, and uncertainty of brain electrical signals [7]. A series of studies on the use of entropy features to characterize EEG signal to detect different EEG states of the brain have been carried out at home and abroad. Min et al. [8] used multiple entropy fusion analysis to capture four important channel regions based on electrode weights. Four classifiers are combined together to construct an evaluation model for detecting driver fatigue state, and finally, obtained a classification accuracy of 98.3%. Ye [9] proposed a driving fatigue state recognition method based on sample entropy, t-test, and kernel principal component analysis. Fatigue driving data was classified and its recognition accuracy rate was 99.27%. Vladimir et al. [10] explored the multi-scale brain signal complexity of the EEG spectrum and the changes of the power law scaling index, finding that nonlinear dynamical properties of brain signals accounted for a smaller portion of entropy changes, especially in stage 2 sleep. Zou et al. [11] proposed a method based on the empirical mode decomposition (EMD) of multi-scale entropy on the recorded forehead EEG signals to recognize fatigue driving. Results indicated that the classification recognition rate of EMD multi-scale fuzzy entropy feature is up to 88.74%, which is 23.88% higher than single-scale fuzzy entropy and 5.56% higher than multi-scale fuzzy entropy. The above valuable studies show that entropy can well characterize the complexity, non-linearity, and uncertainty of EEG signals. Due to the characteristics of entropy, the FBN constructed using entropy features can more accurately identify the brain states. Due to the threshold selected by the FBN through the sliding window which has a certain time-domain resolution, the accuracy of the FBN constructed using entropy is more stable when the EEG signal classification of different time intervals is intercepted.

Therefore, an EEG state recognition model of an FBN based on entropy features is proposed in this paper. By testing the entropy-based FBN model on the real data set of fatigue driving, an FBN fatigue driving state recognition model based on fuzzy entropy (FE) [12] (abbreviated as FE_FBN) is constructed. In this paper, fuzzy entropy, sample entropy (SE) [13,14], approximate entropy (AE) [15], and spectral entropy (SPE) [16] are used to calculate the original signal; mutual information (MI) [17,18], pearson correlation coefficient (PEA) [18,19], and correntropy coefficient (CORE) [18,20] are used to measure the correlation among electrodes; and then six kinds of classifiers are chosen in the experimental environment to find the most suitable classifier. Finally, entropy, correlation coefficient, and the appropriate classifier are combined to construct the fatigue driving state recognition model. The FE_FBN has the best performance in fatigue driving state recognition. On the same data set, even if the lengths of the intercepted EEG signals are different, a higher accuracy and more stable classification results can be obtained than that of the model proposed by Ye [9].

## 2. Materials and Methods

### 2.1. Entropy-Based FBN Model Architecture

The FBN models based on FE, SE, AE, and SPE (abbreviated as FE/SE/AE/SPE_FBN), are collectively referred to as the entropy-based FBN model (abbreviated as EN_FBN). The model structure and components of EN_FBN are illustrated in this section. A brief description of the model structure is shown in Figure 1.

### 2.2. Implementation Method of FBN Model Based on Entropy

#### 2.2.1. Entropy Feature Calculation

FE, SE, AE, and SPE are calculated on the original data. AE was proposed by Pincus et al. [15]. It is a nonlinear dynamic parameter used to quantify the regularity and unpredictability of time series fluctuations, which uses a non-negative number to represent the complexity of a time series and reflects the possibility of new information in the time series. The more complex the time series, the greater the approximate entropy. Its calculation method is mentioned in the literature [15]. SE is an improvement of approximate entropy and its calculation method does not depend on the data length, which was proposed by Richman et al. [13]. It is also a measurement method of time series complexity and analysis of a nonlinear signal. Its calculation method is mentioned in the literature [14]. SPE describes the relationship between power spectrum and entropy rate. In this paper, the normalized Shannon entropy is used to evaluate SPE [16]. Its calculation method is mentioned in the literature [16]. FE was first proposed by Chen et al. [12]. Due to the best classification performance being on the model of FE_FBN, its calculation method is shown below:•Given a N-dimensional time series [u(1),u(2),…,u(N)], and define phase space dimensions m(m≤N−2) and similarity tolerance r(r=0.2×std), reconstruct phase space:
(1)$$X\left(i\right)=\left[u\left(i\right),u\left(i+1\right),\ldots ,u\left(i+m-1\right)\right]-u_{o}\left(i\right),i=1,2,\ldots ,N-m+1\;$$
where $u_{0}\left(i\right)$ is $u_{0}\left(i\right)=\frac{1}{m}\sum_{j=0}^{m-1}u\left(i+j\right)$;•Fuzzy membership function is introduced as:
(2)$$A\left(x\right)=\left\{\begin{array}{c} 1,x=0\\ exp(\frac{-{\left(x\right)}^{2}}{r}),x>0 \end{array}\right.\;$$
where *r* is the similarity tolerance. Calculate i=1,2,…,N−m+1 as:
(3)$$A_{ij}^{m}=exp\left(\frac{{\left(-d_{ij}^{m}\right)}^{2}}{r}\right),j=1,2,\ldots ,N-m+1,\;\mathrm{and}\;j\neq i\;$$•Where $d_{ij}^{m}$ is the maximum absolute distance between the window vectors X(i) and X(j), calculated as:
(4)$$d_{ij}^{m}=d\left[X\left(i\right),X\left(j\right)\right]=max_{p=1,2,\ldots ,m}(|u\left(i+p-1\right)-u_{0}\left(i\right)|-|u\left(j+p-1\right)-u_{0}\left(j\right)\left|\right)\;$$•After calculating the average for each *i*, the following formula can be obtained:
(5)$$C_{i}^{m}=\frac{1}{N-m}\sum_{j=1,j\neq i}^{N-m+1}A_{ij}^{m}\;$$•Define:
(6)$$\Phi^{m}\left(r\right)=\frac{1}{N-m+1}\sum_{j=1,j\neq i}^{N-m+1}\log C_{i}^{m}\left(r\right)\;$$•The fuzzy entropy formula of the original time series is:
(7)$$FuzzyEn\left(m,r,N\right)=\lim_{N\to \infty }\left[\ln\left(\Phi^{m}\left(r\right)-\Phi^{m-1}\left(r\right)\right)\right]\;$$For a finite data set, the fuzzy entropy formula is:
(8)$$FuzzyEn\left(m,r,N\right)=\ln(\Phi^{m}\left(r\right)-\Phi^{m-1}\left(r\right))\;$$

In this paper, r=0.2×std, where *std* = 1.25, *r* = 0.25, are described specifically in the paper of Ye [9].

#### 2.2.2. The Required Method of FBN Construction


(1)Synchronization correlation coefficient


MI, which was proposed by Kraskov et al. [17], is a quantity used to measure the correlation between two pieces of information and can be used to analyze the information flow between two systems or between the constituent subsystems of a complex system. Its calculation formula is as shown below.

Let *X* and *Y* be random variables with probability density functions. p(x)=PX=x, p(y)=PY=y. The mutual information between *X* and *Y* is:(9)$$I\left(X,Y\right)=\sum_{x}\sum_{y}p\left(x,y\right)log\frac{p(x,y)}{p(x)p(y)}\;$$

PEA was proposed by Pearson et al. [19]. Since the channel based on the EEG data is simultaneous and the entropy property is fixed, PEA can be used to calculate the strong-weak connection between different electrodes. Its calculation formula is as shown below:(10)$$R_{xy}=\frac{N\cdot \sum_{i=1,j=1}^{N}x_{i}y_{j}-\left(\sum_{i=1}^{N}x_{i}\right)\left(\sum_{j=1}^{N}y_{j}\right)}{\sqrt{N\cdot \sum_{i=1}^{N}x_{i}^{2}-{\left(\sum_{i=1}^{N}x_{i}\right)}^{2}}\sqrt{N\cdot \sum_{j=1}^{N}y_{j}^{2}-{\left(\sum_{j=1}^{N}y_{j}\right)}^{2}}}\;$$

CORE is an extension of the correlation coefficient, which performs well on high-order statistical sum, or nonlinear relationship signals. In this paper, CORE are calculated for the entropy properties between different electrodes. The calculation formula is as shown below:•Define the CORE of random variables *X* and *Y* as: $V_{\delta }\left(X,Y\right)=E\left[k_{\delta }\left(X,Y\right)\right]$, where *E* represents the expectation operator, $k_{\delta }\left(\cdot \right)$ represents the kernel function, and δ>0 is the kernel width. The Gaussian kernel is usually selected as the kernel function:
(11)$$k_{\delta }\left(x-y\right)=G_{\delta }\left(x-y\right)=\frac{1}{\sqrt{2\pi }\delta }exp\left(-\frac{{\left(x-y\right)}^{2}}{2\delta^{2}}\right)\;$$The selection criteria of the kernel function is very strict, and the selection of δ is based on Silverman’s rule of thumb [21]: $\delta =0.9AN^{-\frac{1}{5}}$, where *A* is the minimum value of the data standard deviation, and *N* is the number of data samples.•Assuming that the joint distribution function of random variables *X* and *Y* is expressed as $F_{xy}\left(x,y\right)$, the CORE is expressed as: $V_{\delta }\left(X,Y\right)=\int G_{\delta }\left(x-y\right)dF_{xy}\left(x,y\right)$. For the limited amount of data and the joint distribution $F_{xy}$ function is unknown, the CORE can be estimated by averaging two finite samples:
(12)$$V_{\delta }^{{}^{\prime }}=\frac{1}{N}\sum_{i=1}^{N}G_{\delta }\left(e_{i}\right)\;$$
where $e_{i}=x_{i}-y_{i},{\left(x_{i}-y_{i}\right)}_{i=1}^{N}$ are *N* sampling points of the joint distribution function $F_{xy}$.(2)Threshold selection

In paper [22], Guo has proved that FBN has the best classification recognition if the network sparsity is between 8% and 32%, and take 1% as the step size of threshold selection [22]. Threshold selection of our model is based on his theory. On the other word, the best threshold of the network sparsity between 8% and 32% is obtained based on the classification recognition rate.(3)Network measurement

Three classically representative network measurements are used in the experiment, the name of which are average path length (APL) [22], clustering coefficient (CC) [22], and local efficiency (LE) [22].

#### 2.2.3. Verification Standard of “Small World” Property of Network

Watts et al. referred to a network with the high clustering coefficient and the shortest path length as a “small world” network [23], and Humphries et al. believed the networks that conform to indicator σ=γ/λ>1 are “small world” networks, where $\gamma =C_{p}/C_{p}^{rand},\lambda =L_{p}/L_{p}^{rand}$. The bigger of σ, the stronger of “small world” property [24].

In this paper, $C_{p}$ represents the coefficient of polymerization of the model, $C_{p}^{rand}$ is the coefficient of polymerization of the ER random network, which is obtained by the arithmetic averaging of the clustering coefficients of all nodes in the network [22]. $L_{p}$ refers to the characteristic path length of the model, $L_{p}^{rand}$ is the characteristic path length of the ER random network, which is the average of the shortest path length between any two nodes in the network [22].

The ER random network, proposed by Erdos and Renyi [25], is a simple graph without repeated edges and loops. The basic idea of the model is to connect each pair of *N* nodes with probability *p* to generate an ER random network. The ER random network used in this paper has 30 nodes corresponding to 30 electrodes. The points are connected with a probability of 0.1. The specific generated graph is shown in Section 3.2.

#### 2.2.4. Classifer

The classifiers artificial neural network (ANN), decision tree (DT), random forest (RF), k-nearest neighbor (KNN), adaboost (AD), and support vector machine (SVM) are used. For each classifier, the test result of 10-fold cross-validation is used as the classification accuracy.

### 2.3. The Model Framework Construction Flow of EN_FBN

In order to allow readers to understand our model step by step, a brief introduction to the model construction is given instead of a detailed description of carrying specific data. The detailed description is shown in Section 2.4.3. The organizational framework of the model structure is shown in Figure 2.

•Calculate entropy under different fatigue driving states in S seconds of R individuals and construct the matrix. Suppose the entropy is $E_{\left(S\times R)\times l\right)}$. (S×R) stands for the size of row of *E*. *l* stands for size of column of *E* and the electrode numbers;•Construct synchronization correlation coefficient matrix. The adjacent matrix is assumed to be $C_{m\times n}$, where *m* and *n* stand for rows and columns of *C*, and *n* represents the electrode numbers;•Construct the model EN_FBN;•Extract the network measurement matrix as the feature matrix. The network measurement matrix is assumed to be $M_{i\times j}$, where *i* and *j* stand for rows and columns of *M*, and *j* represents the electrode numbers;•Put the feature matrix into classifier and get the test result through 10-fold cross-validation.

### 2.4. Data Matrix Construction and EN_FBN Model Construction Algorithm Based on the Real Data Set of Fatigue Driving

#### 2.4.1. Experiment Data

The EEG data was provided by Mu’s team [26]. This data set was obtained from 26 volunteers who volunteered to participate in the car simulation driving experiment. Each of them recorded two sets of experimental data—fatigue driving state and non-fatigue driving state data. The current states of the individual’s self-perception were recorded before and after the experiment, in order to understand the volunteers’ mental and fatigue states. During the experiment, at the beginning, EEG data in the current resting state (or we can say EEG data in non-fatigue driving state) of these experimental individuals were recorded under the condition of adequate sleep and regular diet on the previous night. Then these individuals were required to keep driving for 40 min, and then through the professional questionnaire to check the current state of these individuals [26] and record the fatigue EEG data.

The EEG data is a 32-electrode 600-second time series at a sampling rate of 1000 Hz, which are 300 s of fatigue data and 300 s of non-fatigue data, respectively. After collecting the data, Mu et al. used Neuroscan 4.5 to preprocess the collected data. The frequency range of the data was 0.15 Hz to 45 Hz. The main steps of data preprocessing include drift removal, electrooculogram removal, artifact removal, baseline correction, and filtering [26]. In view of the abnormal conditions that may appear in the experimental process, such as sneezing, coughing, being suddenly frightened, and so on, the EEG drift was removed via an artificial method. For the obvious electrooculogram, mainly vertical electrooculogram, the team deleted it. They used transform-artifact rejection to remove artifacts in EEG signals and chose the time-domain (time) according to the experience, which was in the range of ±50 to ±100. For the data that does not appear in the baseline after processing, one linear correction or two baseline correction are usually needed. The main purpose of digital filtering was to get the EEG data of the main frequency band. In this paper, 1.5 Hz to 70 Hz band-pass filter was used. The study met the ethical principles of the Declaration of Helsinki and the current legal requirements. Ethical approval for this work was also obtained from the Academic Ethics Committee of the Jiangxi University of Technology. All individuals gave their informed consent for their inclusion before they participated in the study.

#### 2.4.2. Construction of Data Matrix

Bring the real data set of fatigue driving into the model framework, which is mentioned in Section 2.3, the symbol *S* and *R* are assumed to $S^{{}^{\prime }}$ and $R^{{}^{\prime }}$. $S^{{}^{\prime }}$ seconds EEG signal samples of $R^{{}^{\prime }}$ individuals were taken as an example. Due to the experimental results of different numbers of people and seconds being compared many times, no specific data is given in Section 2. In order to calculate the synchronization correlation coefficient and network measurement, the content of this section was divided into three parts.

In the first part, the construction process of the entropy matrix is described in detail. In the second part, the construction process of the adjacency matrix is described in detail. In the third part, the network measurement of FBN is calculated and the network measurement matrix is constructed to form the feature matrix.(1)Construction of the entropy matrix

Firstly, calculate entropy and construct the entropy matrix. Four entropy data matrices called FE matrix, SE matrix, AE matrix, and SPE matrix are obtained through the constructed method. The data format of each of them is explained as follows: The EEG data of $S^{{}^{\prime }}$ second non-fatigue driving state and $S^{{}^{\prime }}$ second fatigue driving state are intercepted from the data of $R^{{}^{\prime }}$ individuals. Calculate an entropy value within 1000 Hz EEG data per second. Assume that every entropy value is a symbol *e*. The $S^{{}^{\prime }}$ seconds non-fatigue driving state entropy data matrix $E_{JX}$ and the $S^{{}^{\prime }}$ seconds fatigue driving state entropy data matrix $E_{ZD}$ of each entropy are shown in Figure 3. The rows stand for the entropy value of every second in 30 electrodes, and columns stand for the entropy value of every electrode in $S^{{}^{\prime }}$ seconds of $R^{{}^{\prime }}$ individuals in the figure.(2)Construction of adjacent matrix

Secondly, the synchronous correlation coefficients between different electrodes are calculated. Three matrices called MI matrix, PEA matrix, and CORE matrix are obtained for each entropy matrix by this constructed method. The size of each of them is $\left(2\times \left(R^{{}^{\prime }}\times I\times \left(30\times 30\right)\right))\right)$, where 2 means that each individual has two EEG states, and $R{}^{\prime }$ represents the number of individuals participating in the experiment. Most authors construct an FBN with every second of EEG data, which will lead to a lot of calculations and repeated information. In this paper, the intercepted $S^{{}^{\prime }}$ seconds are divided into *I* parts of each individual. Every adjacent matrix will be calculated by the data of entropy matrix within every $S^{{}^{\prime }}$/*I* second, which can reduce the amount of calculation while removing duplicate information. Within $S^{{}^{\prime }}$ seconds, each individual has *I* adjacent matrices. 30×30 represents the correlation coefficient matrix size, which will construct an FBN after the threshold selection. The value of every synchronous correlation coefficient is assumed as a symbol *c*. Two matrices are used to express the synchronous correlation coefficient matrix of $S^{{}^{\prime }}$ seconds of $R^{{}^{\prime }}$ individuals (into fatigue driving state data matrix ($C_{ZD}$) and non-fatigue driving state data matrix ($C_{JX}$)), as shown in Figure 4. The symbol *c* using the example $c_{12}$ means that the synchronous correlation coefficient is based on entropy value in electrode 1 and electrode 2 within $S^{{}^{\prime }}$/*I* seconds.(3)Construction of network measurement matrix

Finally, after selecting the appropriate threshold, the network measurement of the data matrix is calculated and the network measurement matrix is constructed. The size of each network measurement matrix is $\left(25\times \left(2\times \left(R^{{}^{\prime }}\times I\times 30\right)\right)\right)$. A specific explanation is similar with the construction of the adjacent matrix. Assume that the value of each network measurement is a symbol *m*. The adjacency matrix of 30×30 in each row is calculated as one row, and the row size is (1×30). There are 25 pairs of network measurement matrices under non-fatigue state ($M_{JX}$) and fatigue state ($M_{ZD}$) of all experimental individuals, as shown below:

#### 2.4.3. Construction Algorithm of EN_FBN Model Based on the Real Data Set of Fatigue Driving

According to the flow in Section 2.3, the specific algorithm steps used for the real data set of fatigue driving to construct an EN_FBN fatigue driving recognition model are provided. In this section, two algorithms are provided and the EN_FBN model will be constructed accordingly.(1)The first algorithm: Sparse-based FBN algorithm

**Algorithm input:** Synchronous correlation coefficient matrix $C_{m\times n}$ of size M×N, where the network sparsity is d(0<d<1).

**Algorithm output:***d* functional brain networks $g_{k}$ based on entropy.
•The algorithm begins;•Set the threshold minimum value *d* and the maximum value d_max through the method mentioned in Section 2.2.1;•Define the loop invariant $d\leq d_{max}$, and the loop begins;•Calculate the number of edges V of the matrix $C_{m\times n}$, and sort the weights of the edges of the matrix $C_{m\times n}$ from large to small;•Select the sparsity *d*, and generate the number of network edges $V_{1}$ according to the formula $V_{1}=V\times d$;•Reserve the front side $V_{1}$ of the matrix $C_{m\times n}$, and round off the rest (set the corresponding position of the matrix to 0). Then, generate an FBN $g_{k-d}$;•Increase value *d* by the formula d=d+0.01, and compare the sparsity *d* and $d_{max}$. If $d\leq d_{max}$, jump back to the third step to continue the calculation;•If $d>d_{max}$, the loop ends;•The algorithm ends.(2)The second algorithm: EN_FBN construction algorithm•Calculate the entropy features under different fatigue driving states in $S^{{}^{\prime }}$ seconds of $R^{{}^{\prime }}$ individuals (the specific method is mentioned in Section 2.2.1) and construct entropy matrix (the specific method is mentioned in Section 2.4.2). Suppose the entropy is $E_{(S^{{}^{\prime }}\times R^{{}^{\prime }})\times 30}$, where $\left(S^{{}^{\prime }}\times R^{{}^{\prime }}\right)$ stands for the size of row of *E*, and 30 stands for size of column of *E*, which represents the electrode numbers;•Construct the synchronous correlation coefficients matrix based on the matrix $E_{(S^{{}^{\prime }}\times R^{{}^{\prime }})\times 30}$ (the specific method is mentioned in Section 2.2.2) and construct adjacent matrix (the specific method is mentioned in Section 2.4.2). The adjacent matrix is assumed to $C_{(R^{{}^{\prime }}\times I\times 30)\times 30}$, where $\left(R^{{}^{\prime }}\times I\times 30\right)$ stands for the size of row of *C*, and 30 stands for size of column of *C*, which represents the electrode numbers;•Construct the sparse-based FBN model according to the first algorithm;•Construct the network measurement matrix (the specific method is mentioned in Section 2.4.2). The network measurement matrix is assumed to $M_{(R^{{}^{\prime }}\times I)\times 30}$, where $\left(R^{{}^{\prime }}\times I\times 30\right)$ stands for the size of row of *M*, and 30 stands for size of column of M, which represents the electrode numbers;•Input each pair matrix $M_{JX}$ and $M_{ZD}$ to the classifiers proposed in Section 2.2.3, and get the test result through 10-fold cross-validation.

## 3. Results and Discussion

### 3.1. Experiment and Result Analysis of FBN Based on Four Different Entropy

In Section 3.1 to Section 3.1.2, for the selection and analysis method, the fatigue driving data of 26 experimental individuals are used. Since it is reasonable to use 30 s to perform fatigue driving recognition test in practical applications, the model based on the experimental results of 30 s non-fatigue state data and fatigue state data is mainly analyzed. The 30 s data of each individual are divided into five groups, and the method is shown in Section 2.4.2. The available feature matrix size of each network matrix is (25×(260×30)). By putting the specific data into Figure 5, the matrix is as shown in Figure 6.

These test results were obtained by 10-fold cross-validation, and compared from two aspects of recognition rate and stability in Section 3.1. In Section 3.2, it is verified whether the EN_FBN model meets the “small world” property, and the reliability of the model is further confirmed. In Section 3.3, the appropriate threshold is selected. In Section 3.4, the stability is compared between SE_T_KPCA and FE_FBN.

#### 3.1.1. Comparison Test Results of Classification Recognition Rate among FE/AE/SE/SPE_FBN

This section is devoted to analyzing the classification recognition rate of the proposed model and selecting the best method to build the model.

It can be seen from Table 1 and Table 2 that from the perspective of the classifier selection, the FBN model constructed by using the four entropies respectively proposed performs well in DT, RF, and SVM, among which the performance of RF is the best and its classification accuracy is the highest. Therefore, RF is used as the classifier for classification and recognition in following experiments.

From the perspective of method selection, the classification recognition rate of FBN constructed by FE is higher than that of SE, AE, and SPE. In addition, MI is selected as the synchronous correlation coefficient to construct the FBN. As seen from Figure 7, the optimal recognition rate is up to 99.62%.

In Table 1, FE_MI_APL represents FBN constructed by FE, MI, and APL. The other abbreviations in the table have a similar meaning, such as FE_MI_CC, FE_MI_LE, FE_PEA_APL, FE_PEA_CC, FE_PEA_LE, and so on. The abbreviations in Table 2, Table 3, Table 4, Table 5, Table 6, Table 7, Table 8, Table 9, Table 10 and Table 11 are similar to Table 1.

In Figure 7, the abscissa represents the method combined entropy, synchronous correlation coefficients, and classifiers. The ordinate represents the threshold of classification accuracy, and the number on each column represents the specific value of the classification accuracy of each method.

#### 3.1.2. The Stability Test Results of Each Threshold Recognition Rate of FE/SE/AE/SPE_FBN

It has been confirmed that an FBN model constructed by FE and MI obtained the highest classification recognition. In this section, the feasibility of that is further confirmed through the mean and variance of each optional parameter. The test results are shown in Figure 8 and Figure 9 and Table 5 and Table 6.

In Figure 8 and Figure 9, the abscissa represents the network measurement under different correlation coefficients of FBN, and the ordinate represents the threshold of classification recognition rate, the number on each column represents the specific value of the classification accuracy of each method.

According to these figures, the average and the variance of classification recognition rate of FE_FBN on RF perform better than that of SE/AE/SPE_FBN, and the classification recognition rate of FBN constructed by MI also performs better than that of PEA and CORE on all classifiers.

FE describes the fuzzy degree of a fuzzy set. For fuzzy sets, entropy is an important numerical feature of fuzzy variables. It is an important tool for processing fuzzy information, which used to measure the uncertainty of fuzzy variables. Fuzzy sets are used to describe the set classes in which elements cannot be clearly defined as belonging to a given set. Fuzzy variables take values from this fuzzy set with uncertainty. Therefore, in the process of constructing an FBN, it is more appropriate to describe the uncertainty of the data. The network measurement of FE_FBN makes the topology of resting state and fatigue state of the brain network more separable. Therefore, using FE to construct FBN is the best choice.

Due to the size of MI being closely related to the relationship between variables *X* and *Y*. If *X* and *Y* are more closely related, I(X,Y) is greater. The reason why MI performs well on this model is that it can more accurately estimate the synchronization strength. As for PEA and CORE, compared with the MI with less data restriction and wide application range, the PEA, which is more affected by outliers, and its expansion coefficient on the high-order non-statistical sum CORE, perform poorly in entropy.

According to Table 5 and Table 6, the classification accuracy of FE_FBN using clustering coefficient and local efficiency as features at each threshold is more stable than that of the average path length, while the average recognition rate of using the average path length is higher than that of the clustering coefficient and local efficiency.

### 3.2. “Small World” Property Analysis of EN_FBN

Since the EN_FBN model is first proposed, the result of its “small world” property is shown in this section, which shows that the model is credible. The “small world” property test specific method is described in Section 2.2.3. The ER random graph used in this paper is shown in Figure 10. Since MI performs best on EN_FBN, considering the length of the full text, only the “small world” property of the EN_FBN is constructed using MI as the synchronization correlation coefficient displayed.

In this experiment, 26 individuals were selected, and five functional brain networks were constructed for each individual in 30 s. Each entropy has 260 functional brain networks, due to space limitations, only the “small world" property data of one FBN of each entropy are selected for display in each driving state. The displayed results are shown in Figure 11. In Figure 11, the abscissa represents the 25 thresholds of FBN, and the ordinate represents the δ value threshold. From the figure, it can be seen that the δ values of the networks constructed by this model are bigger than 1 in all threshold, and the trend is generally up with the threshold value, which conforms to the “small world” property described in reference [24].

### 3.3. Threshold Selection of FE_FBN

In this section, the appropriate thresholds for the model were chosen. The threshold selection is based on the FE, MI, and RF (abbreviated as FE + MI + RF). The threshold in the highest value set of each combination method is used as the optimal threshold. Table 7, Table 8 and Table 9 show the accuracy of FE_FBN which contains all indications at each threshold. As we can see, the accuracy of the model is the highest when the network sparsity is 11% and 28%, so they can be selected as the best threshold, on which high classification recognition rate is concentrated. What is more, when network sparsity is 28%, the classification recognition rate performs good, which is only 0.37% lower than the highest point of 99.62%.

### 3.4. Stability Comparison between SE_T_KPCA and FE_FBN

In this section, to further verify the stability of this method, the method SE_T_KPCA proposed by Ye [9] and FE_FBN are compared in the same data set from the aspects of the highest classification accuracy, classification accuracy mean, and variance of all threshold. The reason why the algorithm of Ye was chosen is as follows: First of all, the two methods used the same data set for experimentation. The data comes from the fatigue driving EEG data set produced by the Mu team. Secondly, the team of Ye used the EEG signal entropy feature extraction algorithm based on t-test and KPCA. Our model constructs FBN based on the entropy feature of the EEG signal to identify the EEG state. Both of the two studies are based on the entropy characteristics of EEG signals. In order to further verify the stability of the method, comparing the classification accuracy of SE_T_KPCA and FE_FBN under the same data set in 10 s, 20 s, 30 s, 40 s, 50 s, and 60 s, which are suitable for fatigue driving, to verify that the maximum classification recognition rate of FE_FBN model is higher than that of SE_T_KPCA under different seconds. It can be confirmed that our model not only contains the uncertain characteristics of EEG data but also has a more realistic network topology, even if the length of the intercepted EEG signal is different, it can gain a higher accuracy and more stable classification results than other methods on the same data set. The feature that this model is not sensitive to the length of time, which is of great significance in terms of practicality.

In this experiment, in order to explain the effect of the experiment better, the number of experimental individuals was divided into two groups as Ye [9] divided. The experiment based on the data of 10 individuals is called group one, and the experiment based on the data of 15 individuals is called group two.

In order to avoid under-fitting or over-fitting caused by insufficient data, the number of trees in random forest were adjusted. The number of trees in group one is 2, and the number of trees in group two is 4. The experimental results are shown in Table 10 and Table 11.

As seen from these tables, the highest classification recognition rate of FE_FBN model is higher than that of SE_T_KPCA. The mean value and variance of SE_T_KPCA and FE_FBN in the two group under 10 s to 60 s are shown in Table 12. From the table, it can be prove that our model has a higher recognition accuracy, larger classification mean, and smaller variance.

Entropy characterizes the possibility of new information in a time series, and can effectively describe the complexity, non-linearity, and uncertainty of EEG signals. FBN can effectively describe the network topology. The threshold selected by the FBN through the sliding window has a certain time domain resolution, which can study the dynamic characteristics of the synchronization behavior between brain signals. FE_FBN has a certain time domain resolution, which makes the FBN model contain the chaotic characteristics of EEG signals insensitive to external factors (such as the length of the window for intercepting EEG signals) while ensuring high-precision classification, and can better adapt to changes in an EEG signal length. Therefore, regardless of the length of the window used to intercept the EEG signal data, the accuracy of the model changes little, and the accuracy value performs better.

In this regard, Table 13 and Table 14 are used to explain the classification and recognition of the FE_FBN in 10 s to 60 s at each threshold. In these tables, the average and variance of classification recognition rate performance of APL, CC, and LE of FE_FBN at each threshold performed well in 10 s to 60 s. Among them, the mean value of the first group is between 95 to 97% of the population, and the variance is also between 0.00010 to 0.00035; the mean value of the second group is between 97 to 98% of the population, and the variance is between 0.00010 to 0.00020, so the performance is ideal.

Combined with Table 1, Table 2, Table 3, Table 4, Table 5, Table 6, Table 7, Table 8, Table 9, Table 10, Table 11, Table 12, Table 13 and Table 14 and Figure 3, Figure 4, Figure 5, Figure 6, Figure 7, Figure 8, Figure 9, Figure 10 and Figure 11, it is recommended to use FE + MI + CC + RF to construct the FBN model, in which the number of trees can be adjusted according to the data size. It can achieve the ideal effect of a high classification accuracy, large classification mean, and small classification variance under each threshold.

## 4. Conclusions

In this paper, we focus on selecting appropriate entropy features to characterize EEG signals and construct an FBN. On the real data set of fatigue driving, FBN models based on different entropies are constructed to identify the state of fatigue driving. We think the fuzzy idea of FE is more suitable for this model due to the uncertainty of EEG signals. It makes the topology of the resting state and fatigue state of the brain network more separable. Through analyzing network measurement indicators, the experiment shows that the FBN model based on fuzzy entropy can achieve excellent classification recognition rate and good classification stability. In addition, when compared with the other model based on the same data set, our model can obtain a higher accuracy and more stable classification results even if the length of the intercepted EEG signal is different. This means that this model is not sensitive to the length of time which is of great significance in terms of practicality.

However, there are certain deficiencies. Firstly, in the single-person separation experiment, the classification and recognition rate of this model needs to be improved. Secondly, there was a failure to perform a real-time EEG detection application. We look forward to more research in our model.

## Figures and Tables

**Figure 1 entropy-22-01234-f001:**
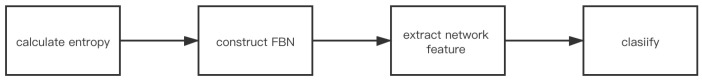
Architecture diagram of the EN_FBN (entropy-based functional brain network) model.

**Figure 2 entropy-22-01234-f002:**
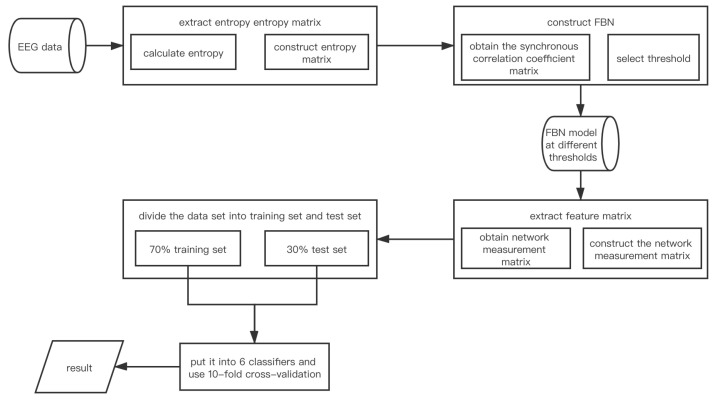
EN_FBN model framewoek.

**Figure 3 entropy-22-01234-f003:**
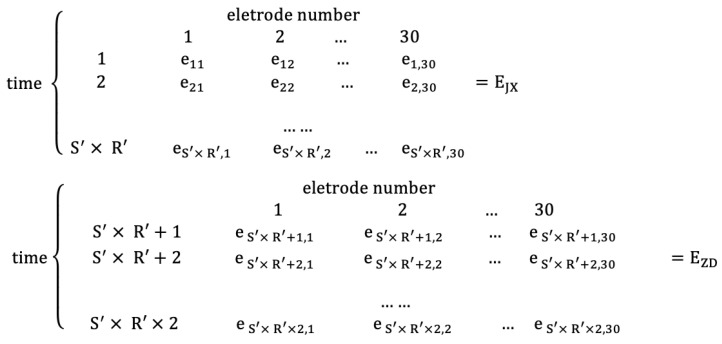
Data matrix of each entropy.

**Figure 4 entropy-22-01234-f004:**
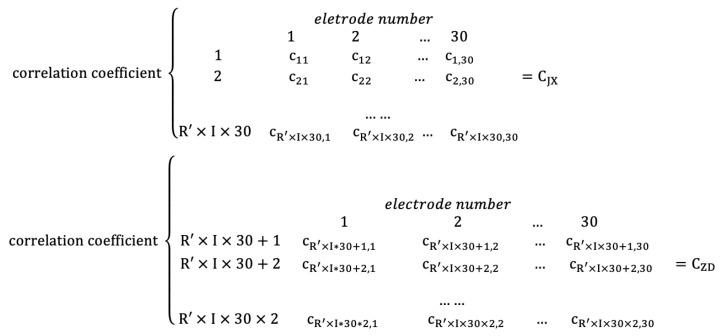
Adjacent matrix of each synchronous correlation coefficient of each entropy matrix.

**Figure 5 entropy-22-01234-f005:**
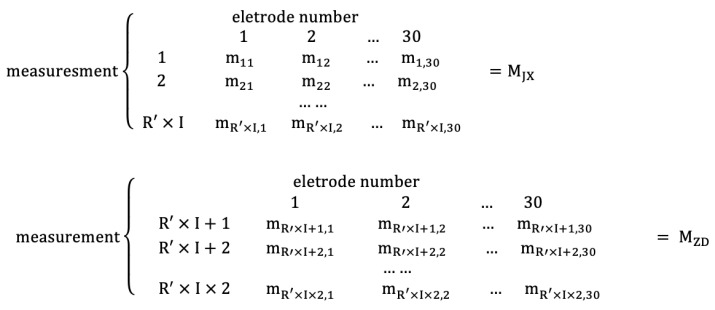
Feature matrix of each network measurement of each adjacent matrix.

**Figure 6 entropy-22-01234-f006:**
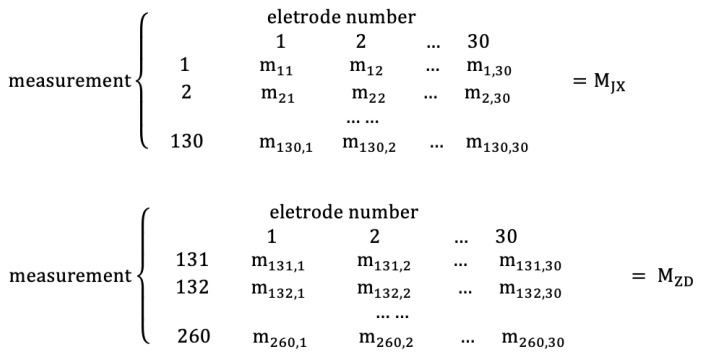
Feature matrix of each network measurement of each adjacent matrix on experimental data.

**Figure 7 entropy-22-01234-f007:**
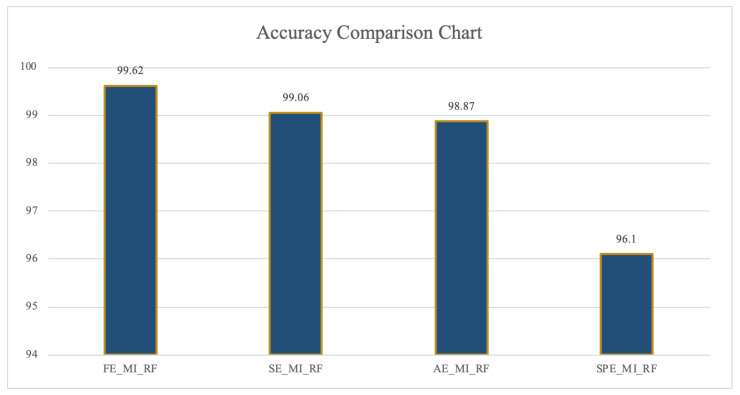
Highest accuracy comparison chart of FE/SE/AE/SpE_FBN (unit: %).

**Figure 8 entropy-22-01234-f008:**
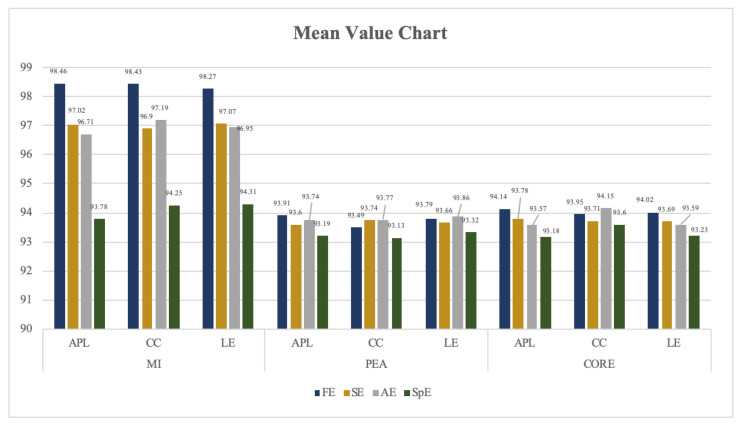
Mean value of classification recognition rate on RF of FE/SE/AE/SPE_FBN (unit: %).

**Figure 9 entropy-22-01234-f009:**
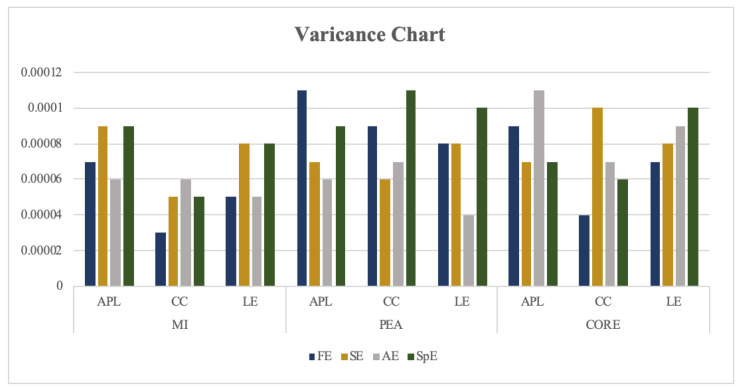
Variance of classification recognition rate on RF of FE/SE/AE/SPE_FBN.

**Figure 10 entropy-22-01234-f010:**
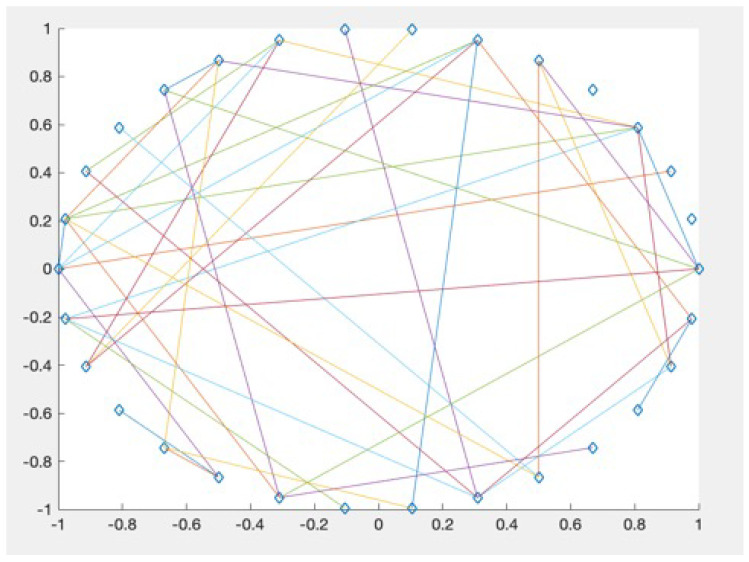
ER random graph.

**Figure 11 entropy-22-01234-f011:**
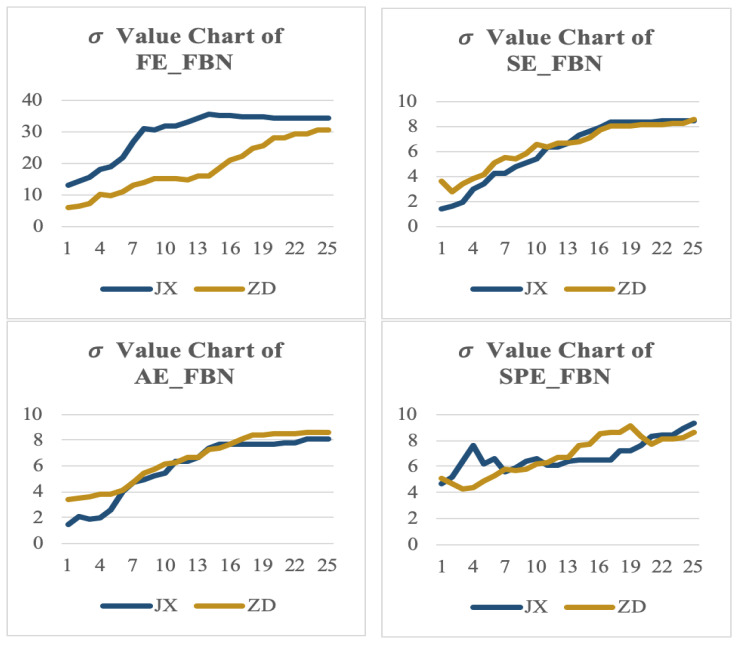
“Small world” property sampling display.

**Table 1 entropy-22-01234-t001:** The highest accuracy of each network measurement of FE_FBN (unit: %). (ANN: artificial neural network; DT: decision tree; RF: random forest; KNN: k-nearest neighbor; AD: adaboost; SVM: support vector machine).

	Classifiers	ANN	DT	RF	KNN	AD	SVM
Feature							
FE_MI_APL		98.36	98.52	99.62	98.97	74.60	93.79
FE_MI_CC		97.74	99.10	99.43	98.44	81.79	98.77
FE_MI_LE		93.74	98.13	99.43	99.12	76.83	98.66
FE_PEA_APL		92.33	93.89	95.53	88.12	86.18	93.96
FE_PEA_CC		89.11	93.85	95.28	88.29	92.13	84.39
FE_PEA_LE		89.41	93.41	95.19	86.65	94.72	88.47
FE_CORE_APL		90.35	94.58	96.04	87.42	86.13	95.53
FE_CORE_CC		90.76	93.89	95.19	87.55	93.38	85.00
FE_CORE_LE		86.94	93.53	96.13	88.06	96.04	88.53

**Table 2 entropy-22-01234-t002:** The highest accuracy of each network measurement of SE_FBN (unit: %, sample entropy (SE)). (ANN: artificial neural network; DT: decision tree; RF: random forest; KNN: k-nearest neighbor; AD: adaboost; SVM: support vector machine).

	Classifiers	ANN	DT	RF	KNN	AD	SVM
Feature							
SE_MI_APL		96.19	97.44	99.06	95.56	72.93	96.02
SE_MI_CC		95.25	96.71	98.30	96.69	80.20	97.20
SE_MI_LE		93.28	96.81	98.21	96.18	94.44	95.91
SE_PEA_APL		90.87	94.53	94.95	87.95	83.37	94.72
SE_PEA_CC		89.34	93.32	95.00	87.96	92.41	86.28
SE_PEA_LE		89.16	94.32	95.19	87.91	95.28	89.07
SE_CORE_APL		91.14	93.69	94.91	87.51	86.43	94.12
SE_CORE_CC		88.72	95.49	96.23	87.61	92.18	84.31
SE_CORE_LE		87.21	93.44	95.00	86.76	94.90	87.53

**Table 3 entropy-22-01234-t003:** The highest accuracy of each network measurement of AE_FBN (unit: %, approximate entropy (AE)). (ANN: artificial neural network; DT: decision tree; RF: random forest; KNN: k-nearest neighbor; AD: adaboost; SVM: support vector machine).

	Classifiers	ANN	DT	RF	KNN	AD	SVM
Feature							
AE_MI_APL		94.81	96.57	97.92	96.13	74.87	95.78
AE_MI_CC		94.63	96.32	98.87	96.32	82.38	96.64
AE_MI_LE		93.01	96.69	98.30	95.72	89.52	95.6
AE_PEA_APL		90.33	94.09	94.81	87.7	85.11	95.09
AE_PEA_CC		90.62	93.08	95.57	88.58	92.29	85.53
AE_PEA_LE		87.17	93.66	95.75	88.33	95.28	87.84
AE_CORE_APL		89.99	93.94	95.94	88.93	83.87	95.09
AE_CORE_CC		89.43	94.25	95.44	87.13	94.09	83.43
AE_CORE_LE		90.26	93.49	95.13	88.21	95.42	87.02

**Table 4 entropy-22-01234-t004:** The highest accuracy of each network measurement of SPE_FBN (unit: %, spectral entropy (SPE)). (ANN: artificial neural network; DT: decision tree; RF: random forest; KNN: k-nearest neighbor; AD: adaboost; SVM: support vector machine).

	Classifiers	ANN	DT	RF	KNN	AD	SVM
Feature							
SPE_MI_APL		90.57	93.47	96.04	88.88	68.44	92.9
SPE_MI_CC		88.08	93.34	95.72	89.24	73.7	91.75
SPE_MI_LE		87.62	94.10	96.10	89.67	87.47	89.47
SPE_PEA_APL		90.00	94.12	95.57	87.67	87.87	93.68
SPE_PEA_CC		89.8	93.42	95.16	88.89	93.64	86.31
SPE_PEA_LE		89.86	93.99	95.38	85.81	95.09	86.8
SPE_CORE_APL		91.18	94.67	94.75	87.01	85.61	94.62
SPE_CORE_CC		93.02	93.22	95.44	88.17	92.15	85.39
SPE_CORE_LE		86.93	93.18	95.47	87.36	94.99	87.86

**Table 5 entropy-22-01234-t005:** Classification precision variance of the network measurement matrix at each threshold of FE_FBN. (ANN: artificial neural network; DT: decision tree; RF: random forest; KNN: k-nearest neighbor; AD: adaboost; SVM: support vector machine).

	Classifiers	ANN	DT	RF	KNN	AD	SVM
Feature							
FE_MI_APL		0.00245	0.00009	0.00007	0.00018	0.00219	0.00091
FE_MI_CC		0.00059	0.00008	0.00003	0.00009	0.00755	0.00173
FE_MI_LE		0.00064	0.00007	0.00005	0.00023	0.00726	0.00553

**Table 6 entropy-22-01234-t006:** Mean value of the classification accuracy of the network measurement matrix at each threshold of FE_FBN (unit: %). (ANN: artificial neural network; DT: decision tree; RF: random forest; KNN: k-nearest neighbor; AD: adaboost; SVM: support vector machine).

	Classifiers	ANN	DT	RF	KNN	AD	SVM
Feature							
FE_MI_APL		92.11	97.15	98.46	96.84	65.86	87.87
FE_MI_CC		93.58	97.15	98.43	96.45	60.97	92.42
FE_MI_LE		90.45	96.84	98.27	96.07	58.72	87.51

**Table 7 entropy-22-01234-t007:** The accuracy of FE + MI + RF at threshold 8% to 16% (unit: %).

	Threshold	1 (8%)	2 (9%)	3 (10%)	4 (11%)	5 (12%)	6 (13%)	7 (14%)	8 (15%)	9 (16%)
Feature										
APL		98.23	98.66	98.11	99.62	97.17	96.49	96.67	97.92	99.12
CC		98.87	98.23	99.25	98.3	98.49	98.65	97.07	98.48	98.53
LE		98.87	98.23	98.11	98.87	99.06	97.32	97.26	99.12	98.34

**Table 8 entropy-22-01234-t008:** The accuracy of FE + MI + RF at threshold 17% to 24% (unit: %).

	Threshold	10 (17%)	11 (18%)	12 (19%)	13 (20%)	14 (21%)	15 (22%)	16 (23%)	17 (24%)
Feature									
APL		98.93	99.25	98.49	97.33	97.92	99.43	99.06	98.74
CC		98.62	98.68	97.95	97.54	98.23	98.3	98.3	98.96
LE		97.78	98.58	99.06	98.11	97.66	98.68	98.02	98.36

**Table 9 entropy-22-01234-t009:** The accuracy of FE + MI + RF at threshold 25% to 32% (unit: %).

	Threshold	18 (25%)	19 (26%)	20 (27%)	21 (28%)	22 (29%)	23 (30%)	24 (31%)	25 (32%)
Feature									
APL		99.25	98.84	98.96	99.25	98.87	98.96	98.68	97.92
CC		97.85	98.21	98.3	99.43	98.33	99.06	98.34	98.87
LE		98.2	97.74	98.01	99.43	97.74	98.3	97.31	98.68

**Table 10 entropy-22-01234-t010:** Precision comparison table of SE_T_KPCA and FE_FBN in different seconds (group: group one, unit: %). (LDA: linear discriminant analysis; RF: random forest).

	Method	SE_T_KPCA	FE_MI_APL	FE_MI_CC	FE_MI_LE
Second		LDA	RF, tree=2	RF, tree=2	RF, tree=2
10 s		75.61	97.57	97.96	98.10
20 s		85.19	97.98	98.50	98.33
30 s		99.27	99.39	98.97	99.25
40 s		86.96	99.21	98.73	99.33
50 s		90.55	98.92	98.93	97.94
60 s		94.61	98.56	99.10	98.27

**Table 11 entropy-22-01234-t011:** Precision comparison table of SE_T_KPCA and FE_FBN in different seconds (group: group two, unit: %). (LDA: linear discriminant analysis; RF: random forest).

	Method	SE_T_KPCA	FE_MI_APL	FE_MI_CC	FE_MI_LE
Second		LDA	RF, tree=4	RF, tree=4	RF, tree=4
10 s		80.33	98.60	99.52	99.03
20 s		87,60	99.41	99.19	99.41
30 s		85.08	99.19	99.68	99.31
40 s		90.46	99.35	99.48	99.03
50 s		92.36	99.00	99.35	98.92
60 s		94.74	99.35	99.19	99.52

**Table 12 entropy-22-01234-t012:** Classification accuracy mean value and variance in two groups between SE_T_KPCA and FE_FBN.

	Method	SE_T_KPCA	FE_MI_APL	FE_MI_CC	FE_MI_LE
Group		Mean|Var	Mean|Var	Mean|Var	Mean|Var
**Group one**		88.70%|0.00674	98.61%|0.00005	98.70%|0.00002	98.54%|0.00004
**Group two**		88.43%|0.00274	99.15%|0.000009	99.40%|0.000004	99.23%|0.000006

**Table 13 entropy-22-01234-t013:** Mean and variance of FE_FBN (group: group one).

	Measurement	APL	CC	LE
Second		Mean|Var	Mean|Var	Mean|Var
10 s		95.57%|0.00016	95.35%|0.00027	95.30%|0.00021
20 s		96.15%|0.00013	95.94%|0.00033	95.47%|0.00026
30 s		96.39%|0.00016	96.11%|0.00019	96.06%|0.00026
40 s		96.07%|0.00033	95.80%|0.00019	95.73%|0.00025
50 s		96.74%|0.00012	95.95%|0.00017	95.96%|0.00015
60 s		96.23%|0.00016	96.02%|0.00020	96.12%|0.00028

**Table 14 entropy-22-01234-t014:** Mean and variance of FE_FBN (group: group two).

	Measurement	APL	CC	LE
Second		Mean|Var	Mean|Var	Mean|Var
10 s		96.88%|0.00011	97.27%|0.00016	97.11%|0.00015
20 s		97.62%|0.00011	97.54%|0.00010	97.62%|0.00011
30 s		97.49%|0.00016	97.72%|0.00014	97.91%|0.00020
40 s		97.83%|0.00020	97.62%|0.00011	97.58%|0.00011
50 s		97.12%|0.00014	97.62%|0.00001	97.50%|0.00010
60 s		97.62%|0.00014	97.66%|0.00010	97.54%|0.00017

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
