# Peer review of "Construction and Application of Functional Brain Network Based on Entropy"

_entropy, 2020, doi:10.3390/e22111234_

Round 1

Reviewer 1 Report

This paper presents an investigation on functional brain network construction based on different types of entropy features calculated from measured EEG, which include fuzzy entropy, sample entropy, approximate entropy and spectral entropy, together with different choice of the correlation measures such as mutual information, Pearson correlation and correntropy coefficient, with also different network indices including average path length, clustering coefficient and local efficiency, and different classification methods including artificial neural network, decision tree, random forest, k-nearest neighbor, AdaBoost, support vector machine. The combinations are then tested on some EEG data obtained from a car simulation driving experiment, for classifying driving fatigue and non-fatigue states, and the results show that the combination of fuzzy entropy, mutual information, clustering coefficient, and random forest generates the best classification accuracy.

Overall, functional brain network approach is hot and extremely important in many field such as brain and cognitive science, and the construction of functional brain network approach is for sure critical to the relevant research. However, this paper has some serious defects and thus cannot be recommended in its current version.

First of all, the presentation of the manuscript looks far from acceptable. There are numerous grammatical or spelling errors, and unprofessional or imprecise expressions, formats, plus many duplications and redundancies, which makes the reading somewhat difficult and the whole study even unreliable. The authors should check and polish their manuscript carefully and thoroughly before formal submission.

What is more, the references cited seem quite limited and not representative, for instance, among those major references, many from the authors’ own group or university (the relation between this work and others work should be explained clearly, especially with [28]), and many are from Chinese journals, and references [19-25] look not very helpful or not necessary. The literature review sounds superficial and somewhat irrelevant. As a consequence, the innovation of this work is limited: all the methods and techniques used here are not new, and the whole study is a test of all types of combinations. Without convincing explanations and evidences, it is not meaningful and useful as it is just for the current dataset or case, and in an offline mode rather than online in a real scenario. Some truly big issues are never touched, for instance, the individual difference. Also, there are so many parameters or configurations in each part, it is not clear whether already optimized for fair comparison.

Author Response

Responds to the reviewer’s comments:

All the authors of the manuscript once again deeply appreciate the reviewer! Thank you for giving us the most valuable amendments and suggestions. These superb professional comments and suggestions have benefited us a lot. We carefully comprehend all the revision opinions and suggestions you put forward, and have made every effort to revise the manuscript in accordance with your comments. If there is anything wrong, please sincerely give us your guidance. If there is any rashness, we kindly ask you for your understanding and tolerance.

Please see the attachment. The revised manuscript has been uploaded.

Comment 1: First of all, the presentation of the manuscript looks far from acceptable. There are numerous grammatical or spelling errors, and unprofessional or imprecise expressions, formats, plus many duplications and redundancies, which makes the reading somewhat difficult and the whole study even unreliable. The authors should check and polish their manuscript carefully and thoroughly before formal submission.

Response: Thank you for your valuable comments. For these problems, we make two modifications.

Firstly, we asked the alumni of English majors to modify the description of the entire paper. We are sorry to make it difficult for you to read.

Secondly, we changed the logic of the introduction (line 13 to line 71), deleted some unrepresentative references, and added citations of English articles. The revision of the introduction helps to improve the review experts' understanding of the model proposed in this paper.

Comment 2: What is more, the references cited seem quite limited and not representative, for instance, among those major references, many from the authors’ own group or university, many are from Chinese journals, and references [19-25] look not very helpful or not necessary. The literature review sounds superficial and somewhat irrelevant. 

Response: Thank you for your valuable comments. We made the following changes to your question. Firstly, the introduction logic has been changed. Secondly, many of the references you mentioned that are not representative or unnecessary have been deleted. In addition, we cited some representative papers to enhance the relevance of the literature and the topic. The modification in the reference of introduction is in line 13 to line 71. The references, which are cited in the introduction, are revised as follows: “

[1] Liang, X.; Wang, J.H.; He, Y. Human brain connection group research: brain structure network and brain function network[J]. Chinese Science Bulletin 2010, 55(16): 1565-1583.

[2] Chen, L. Research and analysis of brain network based on brain fatigue [D]. Hebei University of Technology 2015.

[3] Ma, H.J. Construction and application of functional brain network based on mutual approximation entropy and mutual sample entropy [D]. Xidian University 2014.

[4] Conrin, S.D.; Zhan, L.; Morrissey, Z.D.; Xing, M.; Forbes, A.; Maki, P.M.; Milad, M.R.; Ajilore, O.; Leow, A. Sex-by-age differences in the resting-state brain connectivity [J]. 2018.

[5] Luo, Y.W. Application of functional brain network based on shortest path tree in fatigue driving state recognition [D]. Nanchang University 2019.

[6] Zhao, C.L.; Zhao, M.; Yang, Y. The Reorganization of Human Brain Network Modulated by Driving Mental Fatigue [J]. IEEE JOURNAL OF BIOMEDICAL AND HEALTH INFORMATICS 2017, 21(3): 743-755.

[7] Rifkin, J.; Howard, T. Entropy: A New World View [M]. Shanghai Translation Publishing House 1987.

[8] Min, J.; Wang, P.; Hu, J. Driver fatigue detection through multiple entropy fusion analysis in an EEG-based system [J]. PLOT ONE 2017, 12(12): e0188756.

[9] Peng, Y.T.; Zou, S.L.; Huang, P.F.; Lin, Z.Q.; Ye, B.G.; Qiu, T.R. An algorithm for extracting entropy features from EEG signals based on t-test and KPCA and its application on driving fatigue state recognition [J]. Entropy 2019, 20(9): 701-717.

[10] Vladimir, M.; Kevin, J.M.; Jack, R.; Kimberly, A.C. Changes in EEG multiscale entropy and power‐law frequency scaling during the human sleep cycle [J]. Human Brain Mapping 2019.

[11] Zou, S.L.; Qiu, T.R.; Huang, P.F.; Bai, X.M.; Liu P. Constructing Multi-scale Entropy Based on the Empirical Mode Decomposition(EMD) and its Application in Recognizing Driving Fatigue [J]. Journal of Neuroscience Methods 2020, 341(108691), (ISSN:0165–0270).

[12] Kumar, Y.; Dewal, M.L.; Anand, R.S. Epileptic seizure detection using DWT based fuzzy approximate entropy and support vector machine [J]. Neurocomputing 2014, 133, 271–279.

[13] Lake, D.E.; Richman, J.S.; Griffin, M.P.; Moorman, J.R. Sample entropy analysis of neonatal heart rate variability [J]. Amer. J. Physiol 2002, 283 : R789 – R797 . 

[14] Zhang, Y.; Luo, M.W.; Luo, Y. Wavelet transform and sample entropy feature extraction methods for EEG signals [J]. CAAI Transactions on Intelligent Systems 2012, 7, 339–344.

[15] Pincus; M.; S. Approximate entropy as a measure of system complexity [J]. Proceedings of the National Academy of Sciences of the United States of America 1991, 88, 2297–2301.

[16] Sun, Y.; Ma, J.H.; Zhang, X.Y. EEG Emotional Recognition Based on Nonlinear Global Features and Spectral Features [J]. Journal of Computer Engineering and Applications 2018, v.54;No.912, 121–126.

[17] Kraskov, A.; Stgbauer, H.; Grassberger P. Estimating mutual information [J]. Physical Review E Statistical Nonlinear & Soft Matter Physics 2004, 69, 066138.

[18] Hanieh, B.; Sean, F.; Azinsadat, J.; Tyler, G.; Kenneth P. Detecting synchrony in EEG: A comparative study of functional connectivity measures [J]. Computers in Biology and Medicine 2019, 105:1-15.

[19]Pearson, K. On the Criterion that a Given System of Deviations from the Probable in the Case of a Correlated System of Variables is Such that it Can be Reasonably Supposed to have Arisen from Random Sampling [M]. Breakthroughs in Statistics. Springer New York 1992.

[20] Gunduz, A.; Principe J.C. Correntropy as a novel measure for nonlinearity tests [J]. Signal Process 2009, 89(1): 14–23.”

Comment 3: The relation between this work and others work should be explained clearly, especially with [28]

Response: Thank you for your valuable comments. We have revised the reference. The reference [28] was changed to [25]. The relationship between the algorithm of Ye [9] and our model was added in line 333 to 338: “ The reason of the algorithm of Ye was chosen is as follows: First of all, we used the same data set for experiments. The data comes from the fatigue driving EEG data set produced by the Mu team. Secondly, Ye uses the EEG signal entropy feature extraction algorithm based on t-test and KPCA. Our model is based on the EEG signal entropy feature to construct a functional brain network EEG state recognition model. Both of our studies are based on the entropy characteristics of EEG signals.” The relationship between the reference [25] and our model is shown in section 3.1(line 168 to line 183): “The EEG signal data analyzed in this paper was provided by the team of Mu [25]. This data set was obtained from 26 volunteers who volunteered to participate in the car simulation driving experiment. Each of them was recorded two sets of experimental data—fatigue driving state and non-fatigue driving state data. The current states of the individual's self-perception were recorded before and after the experiment, in order to understand the volunteers’ mental and fatigue states. During the experiment, at the beginning, the current resting state EEG data (or we can say non-fatigue driving states EEG data) of these experimental individuals were recorded under the condition of adequate sleep and regular diet on the previous night. Then these individuals needed to keep driving for 40 minutes, and then through the professional questionnaire to check the current state of these individuals [25] and record the fatigue EEG data.

The EEG data is a 32-electrode 600-second time series at a sampling rate of 1000 Hz, which are 300 seconds of fatigue data and 300 seconds of non-fatigue data, respectively. After collecting the data, it is filtered and prepossessed to remove eye movement interference and other external interferences [25]. All individuals gave their informed consent for inclusion before they participated in the study. The study was conducted in accordance with the Declaration of Helsinki, and the protocol was supported by the National Natural Science Foundation of China.”

Comment 4: As a consequence, the innovation of this work is limited: all the methods and techniques used here are not new, and the whole study is a test of all types of combinations. Without convincing explanations and evidences, it is not meaningful and useful as it is just for the current dataset or case, and in an offline mode rather than online in a real scenario. 

Response: Thank you for your valuable comments. We added the convincing explanations of our model in four parts.

The first part: We added the explanation sentence in lines 58-59: “This model not only contains the uncertain characteristics of EEG data, but also has a more realistic network topology.”

The second part: We added the explanation paragraph in lines 291-301: “We believe that because of the instability of EEG signals, the fuzzy idea of FE is more suitable for this unstable state. The FBN constructed using FE contains more fuzzy information. Its network measurement is more real because of the existence of these fuzzy information, which makes the topology of resting state and fatigue state of brain network more separable. So we think that use FE to construct FBN is the best choice.

Because the size of MI is closely related to the relationship between variable X and Y. If X and Y are more closely related, I(X, Y) is greater. We believe that the reason why MI performs well on this model is that it can more accurately estimate the synchronization strength. As for PEA and CORE, compared with the MI with less data restriction and wide application range, the PEA, which is more affected by outliers, and its expansion coefficient on high-order non-statistical sum CORE, perform poorly in entropy."

The third part: We added the explanation sentence in lines 343-347: “It can be confirmed that our model not only contains the uncertain characteristics of EEG data but also has a more realistic network topology, even if the length of the intercepted EEG signal is different, it can get higher accuracy and more stable classification results than other methods on the same data set. The feature that this model is not sensitive to the length of time is of great significance in terms of practicality.”

The fourth part: We revised the conclusion to make our model more convincing: “In this paper, we study on how to choose an appropriate entropy to construct FBN when entropy feature is used to characterize EEG signals. A fatigue driving state recognition model of FE_FBN was proposed and constructed. A network measurement of FBN was analyzed. Through the experiments and the comprehensive comparison with the functional brain networks constructed based on many different entropies, the results show that the classification recognition rate is excellent, the classification stability is good and the accuracy variance is small. We believe that because of the instability of EEG signals, the FE which extracts data features with fuzzy idea is more suitable for this unstable state. The FE_FBN contains a lot of fuzzy information. Its network measurement is more real because of the existence of these fuzzy information, and it makes the topology of the resting state and the fatigue state of the brain network more separable. So we think that use FE to construct FBN is the best choice. And experiments have also confirmed the effectiveness of constructing FE_FBN for fatigue driving state recognition, which provides a new idea for constructing function brain networks. The stability test of FE_FBN can confirm that our model not only contains the uncertain characteristics of EEG data but also has a more realistic network topology, even if the length of the intercepted EEG signal is different, it can get higher accuracy and more stable classification results than other methods on the same data set. The feature that this model is not sensitive to the length of time is of great significance in terms of practicality.

In addition, there are certain deficiencies in this paper. Firstly, in the single-person separation experiment, the classification and recognition rate of this model needs to be improved. Secondly, failed to perform real-time EEG detection application. We look forward to more research in our model.”

Comment 5: Some truly big issues are never touched, for instance, the individual difference. Also, there are so many parameters or configurations in each part, it is not clear whether already optimized for fair comparison.

Response: Thank you for your valuable comments. Firstly, the content involved in this paper is the content of the entropy-based FBN construction method. This model not only contains the uncertain characteristics of EEG data, but also has a more realistic network topology. Moreover, compared to other method,this model is not sensitive to the length of time the brain electrical state is intercepted. It is more stable. Secondly, we added the explanation for these parameters or configurations in each part. And there is no extra modification time to add the experiments about the individual difference. We are very sorry for this. If you still think this is necessary after reading this reply, please leave a comment.

Reviewer 2 Report

This study compares four different methods, based on entropy, to obtain a brain functional network (FBN) for EEG data. Definitely, the proposal of the article has scientific importance and a great appeal for readers interested in FBN. However, the article has serious structuring problems in writing so that it is exceedingly difficult to understand the methodology of the work and how the results were found. A complete restructuring of the article would be necessary to make it suitable for publication.

Some specific points:

 1) Methodology and results are not clearly defined in the text. Clearly separate methodology, results, and discussion.

2) The continuous use of references for other sessions, throughout the text, leaves you confused and tired of reading in addition to indicating a poor structure of the argument.

3) the item (Threshold selection) needs to be detailed, as it represents an essential element in obtaining the networks.

4) line 181.Include more information about the time left after removing artifacts and which removal method was used.

5) Figures 3,4 and 5 need to be more detailed so that they can be completely understood.

6) The definition of the “S-second EEG signal samples” concept was not clearly defined.

In short, my evaluation of the results, and consequently conclusions, was compromised since I was not able to fully understand the methodology proposed.

Author Response

Responds to the reviewer’s comments:

Thank you for giving us the most valuable amendments and suggestions. There are some symbols in the reply that cannot be written. The modification instructions document is directly uploaded as a PDF. The full text of the revised manuscript has been sent to the assistant editor Keevil Zhang. 

Reviewer 3 Report

The paper compares several methods for functional brain methods construction and tested for identifying the fatigue driving state in drivers.

Despite the obvious amount of work performed and text written, this paper needs to be thoroughly rewritten to become readable even to a reader that is very familiar with information theory, approximate and related entropies, machine learning, and also had some background in driving fatigue.

 In a present form, the paper is very hard to read, there are many typographical errors and language should be improved. Most importantly, the paper offers no methodology explanation, nor the explanation of the results. If no explanation of implemented methods is given, then it would have been a systematic omission. But the authors selected one method to explain in detail. No criterion for selection is given, but they managed to make errors.

Since there are many comments, it would be almost impossible to separate them into different sections.

References:

Lines 64/65: The ApEn and SampEn are introduced in papers by Pincus and by Richman, respectively, with citations numbering thousands. It is unfair to cite a paper that just apply their work as a source reference.

Lines 94/95: It is almost a blasphemy to quote mutual information, one of the crucial terms in information theory, from the references that implement it, at least not with the words “The mutual information formula is from the literature 16,17”. Someone else proposed mutual information. 

Methods: The authors explained very little.

A__

The implemented procedures: the authors use many procedures, but, for some unknown reason, they explain only one, and even this one erroneously: in lines 86/87 they write “The detailed calculation steps for sample entropy are as follows:”, and then they follow with FuzzyEn explanation. With an error in Equation (8). 

 Examples of unexplained procedures, at least a sentence-two per each method should be OK, but the authors wrote differently:

 The following I found unbelievable: one of the procedural steps is “Take the matrix Mi_j as a feature, input it to the classifier proposed in section 1.2.4, and get the test result through ten-fold cross-validation.” Then we return to section 1.2.4. and find 

“1.2.4. Classifier.  

In this paper, the classifiers,ANN,DT,RF,KNN,AD,SVM,are used.”

This is the entire section 1.2.4! 

A particularly annoying property of this text is the general omission of blank spaces after the special characters like comma, full-stop, brackets … and one of the numerous examples is the previous quotation. 

Instead of an explanation, the authors instruct the reader to find the appropriate explanation about the methods in the references: 

1.2.1. ApEn, SampEn, SpecEn are given as the reference, FuzzyEn is explained. This explanation should go to the appendix, with the (slight) differences between these 4 procedures explained. 

1.2.2.1) “The mutual information formula is from the literature [16,17].”

              “the pearson correlation coefficient … formula is from the literature [17].”

              “Correntropy coefficient … formula from a literature [17,18].”

1.2.2.2) It is not explained what is the purpose of the threshold. The first sentence states the existence of two thresholds but gives then two properties. Probably it should mean that two properties are used to define a threshold? The sentence “the network sparsity (8%-32%) is used as the threshold selection range, and the threshold selection is performed in steps of 1% [25].” Seemingly there is a single threshold, but for the unknown purpose because no threshold was mentioned in the previous text. 

1.2.2.3) “three classically representative network measurements are used, which are average path length (APL) [5], clustering coefficient (CC) [5] and local efficiency(LE) [5].”

1.2.3. Authors should define the high clustering coefficient, define the shortest path length (the network is not introduced yet), define the coefficient of polymerization, define ER random network (define ER), define the characteristic path length of the model (you did not introduce any model yet). What are sigma, gamma, lambda? Why changing citation format in this particular sub-sub-chapter (why Watts et al., Humphries et al.?).

1.2.4. already commented. 

Line 213: “Finally, after selecting the appropriate threshold, the network measurement of the data matrix is calculated (the specific method is mentioned in reference [11]),”

All these examples quoted above should be briefly explained in the Appendix, as already said for Entropy. The differences between the methods should be outlined in the result section, to discuss why some of the methods show better results than others.

B__

Method explanation

Line 126 and the following lines: 

The authors should explain what is Ek*l) , (I assume “)” is a typo), what is k, what is l? Is entropy matrix made separate for 4 entropy measures, and for each subject, and for each signal duration? 

How do you preprocess this matrix? In Xm*n, what is m, what is n?

How do you construct FBN? It can be found in Section 1.3.2, but it should be said that the construction of FBN would be shown there! The statement “Construct FBN” before section 1.3.2. is meaningless.

How do you select the network measurements? How do you preprocess them? What is i and j in matrix Mi*j?

Figure 2 is slightly more informative, but it does not answer the previous questions (correct the typos in Fig. 2). 

Lines 133/134 what is M, what is N? Define network sparsity. 

Line 139 now it is clear what the threshold is, but the reference “Set the threshold … through the method mentioned in section 1.2.1.” (an “is” is missing) cannot be exactly applied, see my previous comments on 1.2.1. 

Line 148: Does “calculate the formula” means “increase 'd '   value”?

Contrary to the algorithm given in 1.3.2 that is well explained, the algorithm in 2 is not applicable, as it references the methods in sections 1.2.1, 1.2.2. and 1.2.4 that are not informative (see previous comments). 

Section 3 reveals what signals would be used for this experiment. It should be placed at the beginning of the paper. The first question is: how did the “volunteers who volunteered to participate” (I think this is a pleonasm) become fatigued within a 600 seconds record? Does anyone become fatigued after 5 minutes? Or the records were not continuous in time? I suppose that the authors of [28] who recorded the data have this information. 

Line 195: What does it mean: “Firstly, the entropy matrix is overlays.”

Line 206: “S-second”. If reading straightforwardly, S-second signal means that the duration of the signal is S seconds. However, I suspect that it is actually Sth second sothe duration of signal is 1 second (1000 samples). If it is realy S-second, that means the signal comprises Sx1000 samples, and that would be too much even to ApEn, not to mention FuzzyEn. FuzzyEn, due to the distance evaluation, is known to be time-consuming. I have no writing reference, but at ESGCO conference some years ago, during the post-presentation discussion, the audience (including me) was highly opposed to FuzzyEn, due to the extreme CPU and real-time usage. The authors should comment on this even for 1000 samples signal lengths. Besides, the parameter “r=0.2xstd” requires std, and reliable estimation of std requires stationary data. This was not mentioned in the paper and should be discussed.

What does it mean, to preprocess the entropy?

Figure 3: Do 30 EEG electrodes have some signs? Does some of the preprocessing stages mean that some of the electrodes are excluded?

Matrices are usually not presented as Figures but as an equation. Add in text what rows and columns stand for, and it should not be “times” but “time”, or a number of seconds. 

Line 204: “N represents the number of participants”. In FuzzyEn explanation in the previous sections, N was the signal length. The signs used for variables should be consistent. 

Line 215 “25 represents the threshold number” If it is the same threshold as in 1.3.2 algorithm, then why is it set to exactly 25 when the algorithm is based on “repeat until”? 

Is matrix I the same as previously calculated matrix E? If not, what it is? What are the dimensions of both the matrices (in real numbers)? All the matrices require a detailed explanation!

Line 206 small s and big S, is there a difference?

Figure 4 should be written as an equation, and much better explained. 

The explanation written in lines 215-218 is repeated from the previous explanation. 

Figures 3, 4, and 5 (i.e. corresponding matrices) are too similar to be all presented. Besides, the matrix elements are all denoted as “x”, so it is not clear what they are. 

The first paragraph of 3.3. is not clear. What five groups? 

The sentence is line 229 is grammatically incorrect. 

The paragraph stating what would be the abscises ad ordinates of the future figures that would appear in the next section is redundant. The explanation of a figure should be coupled with the figure. 

A discussion is needed: what is it in the functioning of random forest that it yields the best results? Why FuzzyEn is the best choice, Why the difference in FuzzyEn distance measure in respect to other entropy makes it the best choice? What occurs in MU to make it better than the others?

C__Other

English language should be extensively improved, as well as the style. There are a lot of typographical errors. 

The first sentence in the abstract (that repeats later in the introduction) is too long. It should be split into shorter sentences with clearer meaning. 

Line 30, sentence “However,  the current research is only based on the original EEG data to build functional brain networks, the  degree of confusion and uncertainty of brain networks can not be fully described in these functional brain networks.” Repeats the term “functional brain networks” three times, and a typo “can not” is included.

Line 65: what does it mean: “… Entropy … is used to calculate the original signal”? I thought that EEG was the original signal?

Line 238: Section starts with a capital letter!

Line 240: “It can be seen from table 2-5” should be "from tables" 

Line 315 "choose a appropriate" is a typo.

Some abbreviations (e.g. ANN) are used well before they were defined, and some abbreviations are not defined at all (e.g. BMM, AR). 

Multiplication sign “*” is used in programming languages and this is a text in English. The multiplication sign should be “x” or “·”. 

There are many other technical, typographical, grammatical errors. 

Author Response

(The authors gave the same response as above.)

Round 2

Reviewer 1 Report

The reviewer very much appreciates the revision as well as the efforts behind made by the authors. Nevertheless, after a rough look, the reviewer felt some serious problems still there as the changes seems somewhat superficial.

Though grammatical mistakes have been drastically reduced in this revision, some presentations are still not professional or inaccurate, such as, “the instability of EEG signals”, “a lot of fuzzy information”, “fuzzy idea”, reflecting some misunderstandings or inaccurate understanding, to list just few for instance. BTW, the Abstract which is extremely important has nothing changed, some statements such as “The FBN model constructed by entropy feature is proposed.” and “… the functional brain network (FBN) model constructed by using the electroencephalograph (EEG) data cannot effectively describe the chaotic degree and uncertainty of the brain network, and the entropy features analysis lacks brain network topology information …” look questionable. The authors should make the story especially their innovations and contributions clear with convincing evidence.

The paper reports some results from a comparison study of FBNs with different settings (entropy, classifiers, etc.), despite little innovation, which may be useful and thus acceptable.

However, the manuscript, though has been significantly improved after a major revision, looks still far from "acceptable".

Particularly, many problems in presentation, which reflect limited understanding of the topic.

Some detailed examples:

- The Abstract must be re-written, the current version is inaccurate, unprofessional, and hollow.

e.g, " the FBN models based on different entropy are compared to identify the fatigue driving state. " The comparisons are not for identification.

What are the "chaotic degree" and "uncertainty" here? What is the "chaos" of the brain network? And "confusion"?

- Still so many Chinese references, which is not a normal case in an international journal.

- Section 2 is not clear and accurate enough, e.g., two algorithms.

- "... the protocol was supported by the National Natural Science Foundation of China".

- Statistical analysis should be performed to conclude the difference between two methods.

There are some other similar problems, which should be thoroughly checked and carefully revised before formal submission.

The reviewer has no idea how it can be further improved, especially in a short period, but the manuscript in its current format is not suitable to appear in a rigorous journal.

To sum up, as in the reviewer’s previous comments, this paper shows a test of all combinations with limited innovation, however it is still meaningful if clearly presented with some reasonable analysis and discussion. The authors should present exactly what have been done, with emphasis on the key ideas and findings, compared with the latest work in the literature, in a clear, logic, and concise manner, using precise and professional language.

Author Response

All the authors of the manuscript once again deeply appreciate the reviewer! Thank you for giving us the most valuable amendments and suggestions. These superb professional comments and suggestions have benefited us a lot. We carefully comprehend all the revision opinions and suggestions you put forward, and have made every effort to revise the manuscript in accordance with your comments. If there is anything wrong, please sincerely give us your guidance. If there is any rashness, we kindly ask you for your understanding and tolerance.

We upload a version with a simple change flag. This version only marked in some places with major changes. The modified contents are underlined in red, and the added contents are underlined in blue.

Comment 1: Though grammatical mistakes have been drastically reduced in this revision, some presentations are still not professional or inaccurate, such as, “the instability of EEG signals”, “a lot of fuzzy information”, “fuzzy idea”, reflecting some misunderstandings or inaccurate understanding, to list just few for instance.

Response: Thank you for your valuable comments. The description of “the instability of EEG signals” in full text was revised to “the complexity, non-linearity and uncertainty of EEG signals” which is more professional and precise. For “a lot of fuzzy information” and “fuzzy idea”, the following interpretation was added in line 304 to line 311: “FE describes the fuzzy degree of a fuzzy set. For fuzzy sets, entropy is an important numerical feature of fuzzy variables. It is an important tool for processing fuzzy information, which used to measure the uncertainty of fuzzy variables. Fuzzy sets are used to describe the set classes in which  elements cannot be clearly defined as belonging to a given set. Fuzzy variables take values from this fuzzy set with uncertainty. Therefore, in the process of constructing an FBN, it is more appropriate to describe the uncertainty of the data. The network measurement of FE\_FBN makes the topology of resting state and fatigue state of brain network more separable. Therefore, use FE to construct FBN is the best choice.”

Comment 2: BTW, the Abstract which is extremely important has nothing changed, some statements such as “The FBN model constructed by entropy feature is proposed.” and “… the functional brain network (FBN) model constructed by using the electroencephalograph (EEG) data cannot effectively describe the chaotic degree and uncertainty of the brain network, and the entropy features analysis lacks brain network topology information …” look questionable. The authors should make the story especially their innovations and contributions clear with convincing evidence.

Response: Thank you for your valuable comments. After reading the abstract carefully, we did find that the abstract was somewhat hollow and not professional enough. The rewrote abstract is shown as below: “Functional brain network (FBN) is the most intuitive expression of the dynamic neural activity interaction between different neurons, neuron clusters or cerebral cortex regions, which can effectively describe the characterization of the brain network topology, dynamic properties and function of brain network. How to build an FBN so that it can characterize the features of the brain network accurately and effectively is a challenging subject. Entropy represents the chaos of a system, which can effectively describe the complexity, non-linearity and uncertainty of brain electrical signals. In the field of fatigue driving research, entropy has been widely studied in this field. FBN is a relatively new research direction in this field, and its research based on fatigue driving EEG data construction method has broad prospects. Therefore, it is of great significance to study the construction of FBN network based on entropy. In view of this research problems, we focus on selecting appropriate entropy features to characterize EEG signals, which is used to construct a FBN. On the real data set of fatigue driving, the FBN models based on different entropies are constructed to identify the fatigue driving state. After verifying its "small world" property, the functional brain networks constructed based on different entropies proved to meet the "small world" property. After analyzing and classifying network measurement metrics, it shows that FBN construction based on fuzzy entropy can achieve the effect of excellent classification recognition rate, good classification stability and small variance of recognition rate. Compared with the different model built on the same data set by other members of the same experimental group, even if the length of the intercepted EEG signal is different, our model can obtain higher accuracy and more stable classification results. Experiments show that the proposed model is effective and feasible, which opens up a new idea for the construction of FBN.”

Comment 3: The Abstract must be re-written, the current version is inaccurate, unprofessional, and hollow. e.g, " the FBN models based on different entropy are compared to identify the fatigue driving state.” The comparisons are not for identification

Response: Thank you for your valuable comments. The revised abstract is shown in the response to comment 2. The sentence “he FBN models based on different entropy are compared to identify the fatigue driving state.” has been revised in line 8 as follows: “the FBN models based on different entropies are constructed to identify the fatigue driving state.”

Comment 4: What are the "chaotic degree" and "uncertainty" here? What is the "chaos" of the brain network? And "confusion"?

Response: Thank you for your valuable comments. In the book of “Electroencephalography” (written by Ernst, N.M and Fernando, L.S.), the description that EEG signals are complex, non-linear and random (or “highly uncertain”) can be gotten. In our paper, "chaotic degree" and the "chaos" of the brain network were revised to “the complexity, non-linearity and uncertainty of EEG signals”.

Comment 5: Still so many Chinese references, which is not a normal case in an international journal.

Response: Thank you for your valuable comments. Firstly, the references of Chen et al. [2], Ma et al.[3] and Luo [5] were deleted. Secondly, Meier et al. [2] and Kabbara et al. [3] were added in line 28 to line 38. One of the references is necessary and cannot be deleted (reference 22). After revising, only 2 references are Chinese referencesin line 24 to 25, line 115 to 118. The corresponding references are shown as follows: 

  1. Liang, X.; Wang, J.H.; He, Y. Human brain connection group research: brain structure network and brain function network[J]. Chinese Science Bulletin .
  2. Guo, H. Analysis and classification of abnormal topological attributes of resting function network in depression [D]. Taiyuan University of Technology 2013.

Comment 6: Section 2 is not clear and accurate enough, e.g., two algorithms.

Response: Thank you for your valuable comments. Firstly, the sentence “The first algorithm: sparse-based FBN algorithm” in section 2.3.1 was moved to section 3.3.1, which is after the data introduction section. This makes the overall algorithm easier to understand in Section 3.3. Section 2.3 is only the overall framework flow of this algorithm, and does not give a detailed description . Secondly, in order to allow readers to understand our model step by step, we give a brief introduction to the model construction instead of a detailed description of carrying specific data in section 2.3. The detailed description is shown in section 3.3, which makes our paper more layered.

Comment 7: "... the protocol was supported by the National Natural Science Foundation of China".

Response: Thank you for your valuable comments. The sentence was revised as the following in line 177 to line 180: “Ethical approval for this work was obtained from Academic Ethics Committee of the Jiangxi University of Technology. All individuals gave their informed consent for inclusion before they participated in the study.”

Comment 8: Statistical analysis should be performed to conclude the difference between two methods.

Response: Thank you for your valuable comments. The conclusion after the statistical analysis between two methods were added in line 374 to line 386. The added content is as follows: “Entropy characterizes the possibility of new information in time series, and can effectively describes the complexity, non-linearity and uncertainty of EEG signals. FBN can effectively describes the network topology. The threshold selected by the FBN through the sliding window has a certain time domain resolution, which can study the dynamic characteristics of the synchronization behavior between brain signals. FE\_FBN has a certain time domain resolution, which makes the FBN model contain the chaotic characteristics of EEG signals insensitive to external factors (such as the length of the window for intercepting EEG signals) while ensuring high-precision classification, and can better adapt to changes in EEG signal length. Therefore, regardless of the length of the window used to intercept the EEG signal data, the accuracy of the model changes little, and the accuracy value performs better.”

Reviewer 2 Report

A significant improvement of the text has been made. Only one of the comments need to be reviewed.

Reply to comment 4.

I don't think the authors understood my question. I asked the authors to include an explanation of how the eye movement, muscle artifacts, network fluctuation, etc, were treated. What methods were used and how much of the signal was reduced or what was the total time of the remaining time series after treatment.

Author Response

All the authors of the manuscript once again deeply appreciate the reviewer! Thank you for giving us the most valuable amendments and suggestions. These superb professional comments and suggestions have benefited us a lot. We carefully comprehend all the revision opinions and suggestions you put forward, and have made every effort to revise the manuscript in accordance with your comments. If there is anything wrong, please sincerely give us your guidance. If there is any rashness, we kindly ask you for your understanding and tolerance.

We uploaded a version with a simple change flag. This version only marked in some places with major changes. The modified contents are underlined in red, and the added contents are underlined in blue.

Comment: I don't think the authors understood my question. I asked the authors to include an explanation of how the eye movement, muscle artifacts, network fluctuation, etc, were treated. What methods were used and how much of the signal was reduced or what was the total time of the remaining time series after treatment.

Response: Thank you for your valuable comments. The description for data processing is provided by team of Mu. It has been added in lines 167 to 177 of the paper: “After collecting the data, Mu et al. used Neuroscan 4.5 to preprocess the collected data. The frequency range of the data is 0.15hz to 45hz. The main steps of data preprocessing include drift removal, electrooculogram removal, artifact removal, baseline correction and filtering [24]. In view of the abnormal conditions that may appear in the experimental process, such as sneezing, coughing, being suddenly frightened and so on, the EEG drift is removed by artificial method. For the obvious electrooculogram, mainly vertical electrooculogram, the team deleted it. They used transform-artifact rejection to remove artifacts in EEG signals, and chose the time-domain(time) according to the experience, which was in the range of 50 to 100. For the data that does not appear in the baseline after processing, one linear correction or two baseline correction are usually needed. The main purpose of digital filtering is to get the EEG data of the main frequency band. In this paper, 1.5hz to 70hz band-pass filter is used.”

Reviewer 3 Report

The paper that came to the review is a paper substantially different from the previous version. It is clear, it is readable and most of the comments are resolved.

The increase of readability might be because the essence of the text is already known (a review of five pages cannot be easily forgotten), or because the old and new sentences are tracked side-by-side. In any case, my feeling is that the authors made a considerable effort to improve their paper and to present their findings in which, as I stated previously, they put “an obvious amount of work”. I hope that my positive review is not a consequence of the existing familiarity with the paper.

Figure captions are now more informative and clarify the existence of similar matrices.

In my opinion, although I am not a native English speaker nor I am an expert in languages, that English still needs another thorough check-up. 

Author Response

All the authors of the manuscript once again deeply appreciate the reviewer! Thank you for giving us the most valuable amendments and suggestions. These superb professional comments and suggestions have benefited us a lot. We carefully comprehend all the revision opinions and suggestions you put forward, and have made every effort to revise the manuscript in accordance with your comments. If there is anything wrong, please sincerely give us your guidance. If there is any rashness, we kindly ask you for your understanding and tolerance.

We uploaded a version with a simple change flag. This version only marked in some places with major changes. The modified contents are underlined in red, and the added contents are underlined in blue.

Comment: The paper that came to the review is a paper substantially different from the previous version. It is clear, it is readable and most of the comments are resolved.

The increase of readability might be because the essence of the text is already known (a review of five pages cannot be easily forgotten), or because the old and new sentences are tracked side-by-side. In any case, my feeling is that the authors made a considerable effort to improve their paper and to present their findings in which, as I stated previously, they put “an obvious amount of work”. I hope that my positive review is not a consequence of the existing familiarity with the paper.

Figure captions are now more informative and clarify the existence of similar matrices.

In my opinion, although I am not a native English speaker nor I am an expert in languages, that English still needs another thorough check-up. 

Response: Thank you for your valuable comments. Thank you very much for your approval of our modification. In this revision, we have tried our best to revise our translation and expression. Thank you for your patient guidance again. If you have any other questions, please leave your comments.

Round 3

Reviewer 1 Report

The second revision made by the authors are fully recognized and appreciated by the reviewer. The manuscript has been indeed apparently improved, however, it is still far from the standard.

Taking the Abstract as an example again. Overall, it is too long, and many introductions there are not necessary. The authors could just present what to do, why and how, with the most significant results. The language looks quite clumsy: “in the field”, “in this field”, and again “in this field”. Grammatical error: “this research problems”. “After analyzing and …” and “Compared with …” are somewhat repeated and thus redundant. Why it is “opens up a new idea for …”? Any evidence for “the most intuitive expression”? What is purpose of the “small world property” statement here?

Some other apparent problems:

Page 2: “out at home and abroad”, “the second stage of sleep”, “the performance is particularly obvious”. Reference [8] is by Ye???

Page 3: “similar tolerance” but later “similarity tolerance”. “Because of the FBN construction based on fuzzy entropy has the best classification performance”. The numberings of the steps and equations are confusing.

As shown above, so many problems are found in the Abstract such a short but crucial paragraph. The reviewer cannot check every detail for the authors. While to guarantee a certain level of quality of the journal, the reviewer again requests a thorough and careful revision on the manuscript, with the help of an experienced researcher in the field. It must be done before the formal publication or acceptance or even more detailed comments on the technical parts.

Author Response

Responds to the reviewer’s comments:

All the authors of the manuscript once again deeply appreciate the reviewer! Thank you for giving us the most valuable amendments and suggestions. These superb professional comments and suggestions have benefited us a lot. We carefully comprehend all the revision opinions and suggestions you put forward, and have made every effort to revise the manuscript in accordance with your comments. If there is anything wrong, please sincerely give us your guidance. If there is any rashness, we kindly ask you for your understanding and tolerance.

We simply mark the parts that need to be modified, and upload the last version of the document with the mark to the web page. The modified manuscript has been uploaded to entropy as a new version. We hope that our manuscript can meet your standards. Please see the attachment.

Comment 1: The second revision made by the authors are fully recognized and appreciated by the reviewer. The manuscript has been indeed apparently improved, however, it is still far from the standard.

Response: Thank you for your valuable comments. We are honored to have your appreciation and recognition. We have seriously improved the series of questions you raised and hope to present the manuscript in a better state. Thank you for your guidance and help in revising the manuscript.

Comment 2: Taking the Abstract as an example again. Overall, it is too long, and many introductions there are not necessary. The authors could just present what to do, why and how, with the most significant results.

Response: Thank you for your valuable comments. After this modification, we have controlled the content of the Abstract to less than 200 words. The revised abstract is shown on lines 1 to 14.

Comment 3: The language looks quite clumsy: “in the field”, “in this field”, and again “in this field”. Grammatical error: “this research problems”. “After analyzing and …” and “Compared with …” are somewhat repeated and thus redundant. Why it is “opens up a new idea for …”? Any evidence for “the most intuitive expression”? What is purpose of the “small world property” statement here?

Response: Thank you for your valuable comments. In response to the language and logic issues raised by the reviewers, we made further modifications. Firstly, we deleted and modified these clumsy and wrong language expressions. Secondly, some logic errors have been modified. The specific content is as follows: “Functional brain network (FBN) is an intuitive expression of the dynamic neural activity interaction between different neurons, neuron clusters or cerebral cortex regions. It can characterize the brain network topology and dynamic properties. How to build an FBN to characterize the features of the brain network accurately and effectively is a challenging subject. Entropy can effectively describe the complexity, non-linearity and uncertainty of EEG signals. As a relatively new research direction, the research of the FBN construction method based on EEG data of fatigue driving has broad prospects. Therefore, it is of great significance to study the entropy-based FBN construction. We focus on selecting appropriate entropy features to characterize EEG signals and construct an FBN. On the real data set of fatigue driving, FBN models based on different entropies are constructed to identify the state of fatigue driving. Through analyzing network measurement indicators, experiment shows that the FBN model based on fuzzy entropy can achieve excellent classification recognition rate and good classification stability. Also, compared with the other model based on the same data set, our model can obtain higher accuracy and more stable classification results even if the length of the intercepted EEG signal is different.”

Comment 4: Some other apparent problems: Page 2: “out at home and abroad”, “the second stage of sleep”, “the performance is particularly obvious”. Reference [8] is by Ye???

Response: Thank you for your valuable comments. Firstly, we are sorry for the misunderstanding about the sentence in line 50, which means “...has been carried out at home and abroad.”, but not “out at home and abroad”. We sincerely apologize again for the misunderstanding caused by our writing. Secondly, in lines 56 to 58, “Vladimir et al. [9] explored the multi-scale brain signal complexity of the EEG spectrum and the changes of the power law scaling index, and found that the nonlinear dynamic properties of the brain signal accounted for a small part of the change in entropy. During the second stage of sleep, the performance is particularly obvious.” has been change to “Vladimir et al. [10] explored the multi-scale brain signal complexity of the EEG spectrum and the changes of the power law scaling index, found that nonlinear dynamical properties of brain signals accounted for a smaller portion of entropy changes, especially in stage 2 sleep.”. Thirdly, there is an error in the References. We have modified the References. The revised content is as follows: "[9] Ye, B.G. Research on Recognition Method of Fatigue Driving State Based on KPCA [D]. Nanchang University 2019 .". After adding this reference, there are two Chinese-language references in this manuscript. Finally, we have added a new reference and added the corresponding content in the introduction in lines 30 to 37. The added Reference is in lines 426 to 428. If you have any questions, please contact us.

Comment 5: Page 3: “similar tolerance” but later “similarity tolerance”. “Because of the FBN construction based on fuzzy entropy has the best classification performance”. The numberings of the steps and equations are confusing.

Response: Thank you for your valuable comments. The proprietary phrase "similar tolerance" in the manuscript was changed to "similarity tolerance". And then, the sentence “Because of the FBN construction based on fuzzy entropy has the best classification performance…” is changed to “Because of the best classification performance is on the model of FE_FBN, its calculation method is shown below:” in line 98 to line 99. Finally, we revised the numberings of the steps in the manuscript to the symbols specified in the Instructions for Authors provided by Entropy.

Comment 6: As shown above, so many problems are found in the Abstract such a short but crucial paragraph. The reviewer cannot check every detail for the authors. While to guarantee a certain level of quality of the journal, the reviewer again requests a thorough and careful revision on the manuscript, with the help of an experienced researcher in the field. It must be done before the formal publication or acceptance or even more detailed comments on the technical parts.

Response: Thank you for your valuable comments. We are sorry that there are so many problems with our manuscript. With the efforts of all authors, all sentences in the manuscript have been revised sentence by sentence. The details of the specific changes will be reflected in the uploaded document. Thank you again for your patience.

Reviewer 2 Report

I believe that the proposed issues have been solved. The article can be published in its current form.

Author Response

All the authors of the manuscript once again deeply appreciate the reviewer! Thank
you for giving us the most valuable amendments and suggestions. These superb
professional comments and suggestions have benefited us a lot. We simply mark the
parts that need to be modified, and upload the last version of the document with the
mark to the web page. The modified manuscript has been uploaded to entropy as a new version. Thank you again for your patient guidance of our manuscript.

Reviewer 3 Report

The paper is further improved since the previous version. The English language is improved as well, especially in the abstract. This improvement declines towards the end of the paper. My suggestion is that the Authors should be advised to take the assistance of Entropy language support service.

Round 4

Reviewer 1 Report

The reviewer would like to thank the authors again for their responses and work behind all three revisions. The reviewer further recommends the manuscript to the journal as it contains some findings which could be helpful and interesting to the readers.

On the other hand, though improved every time, to be honest, unfortunately the manuscript looks still not good enough in presentation. The authors are strongly suggested to consider some Proofreading Service, in order to have a significant and thorough improvement. Meanwhile, the authors should be more careful and serious in writing and revising their manuscript.

BTW, the information is not correct for reference

16. Sun, Y.; Ma, J.H.; Zhang, X.Y. EEG Emotional Recognition Based on Nonlinear Global Features and Spectral Features [J]. Journal of Computer Engineering and Applications 2018, v.54;No.912, 121–126.

It is in Chinese and should be

SUN Ying, MA Jianghe, ZHANG Xueying. EEG emotion recognition based on nonlinear global features and spectral feature. CEA, 2018, 54(17): 116-121.